# The role of hesitancy and infrastructure in the equity and efficiency of COVID-19 vaccine administration

**Noah Chicoine**[1]ᵒ*, **Noah Schnipper**[2]ᵒ, **Jacqueline Griffin**[1]

**1** Department of Mechanical and Industrial Engineering, Northeastern University, Boston, MA, United States of America, **2** Division of Engineering, Lafayette College, Easton, PA, United States of America

ᵒ These authors contributed equally to this work.

* chicoine.n@northeastern.edu

**Data Availability Statement:** All relevant data are within the paper and Supporting information files.

**Funding:** The author(s) received no specific funding for this work.

## Abstract

After the first COVID-19 vaccines received emergency use authorization from the U.S. FDA in December 2020, U.S. states employed vaccine eligibility and administration plans (VEAPs) that determined when subgroups of residents would become eligible to receive the vaccine while the vaccine supply was still limited. During the implementation of these plans, public concern grew over whether the VEAPs and vaccine allocations from the federal government were resulting in an equitable and efficient vaccine distribution. In this study, we collected data on five states' VEAPs, federal vaccine allocations, vaccine administration, and vaccine hesitancy to assess the equity of vaccine access and vaccine administration efficiency that manifested during the campaign. Our results suggest that residents in states which opened eligibility to the vaccine sooner had more competition among residents to receive the vaccine than occurred in other states. Regardless of states' VEAPs, there was a consistent inefficiency in vaccine administration among all five states that could be attributed to both state and federal infrastructure deficits. A closer examination revealed a misalignment between federal vaccine allocations and the total eligible population in the states throughout the campaign, even when accounting for hesitancy. We conclude that in order to maximize the efficiency of future mass-vaccination campaigns, the federal and state governments should design adaptable allocation policies and eligibility plans that better match the true, real-time supply and demand for vaccines by accounting for vaccine hesitancy and manufacturing capacity. Further, we discuss the challenges of implementing such strategies.

## Introduction

In early 2021, as United States residents and others around the world started to receive vaccines for COVID-19, public concern grew over whether the limited number of vaccines were being allocated fairly within the United States. News sources, such as the Washington Post and the New York Times, brought this concern to the general public by highlighting historical instances of unfair vaccine allocation during pandemics [1, 2]. The National Academies

**Competing interests:** The authors have declared that no competing interests exist.

Committee on Science, Engineering, and Medicine also emphasized the need to provide sufficient vaccination infrastructure, especially for communities disproportionately affected by COVID-19 (mostly racial and ethnic minority groups), as they already experience health inequities propagated by structural and systemic factors [3]. These concerns were warranted, as Heier Stamm et. al. (2017) demonstrated that during the H1N1 vaccination campaign, differences in population density and healthcare infrastructure in the southern United States resulted in local inequities in vaccine access [4].

As more of the general public became eligible to receive the vaccine, concerns regarding the equity of vaccine access continued to grow [5]. In later months, access to vaccines was not only dependent on government allocations to the states, but was also determined by individual state's *vaccine eligibility and administration plans* (VEAPs), their population's hesitancy towards the vaccine, and the state's vaccination administration infrastructure.

States' VEAPs varied considerably. For example, certain states opened universal eligibility to the vaccine months sooner than others, despite the fact that state vaccine allocations were still small at the time. Other states' VEAPs opened eligibility much slower, attempting to ensure that the most high risk individuals received vaccines first [6, 7].

Additionally, some states had more residents who were hesitant to receive the vaccine, and elected not to get it, essentially reducing the true demand for vaccines. Meanwhile, other states had lower vaccine hesitancy rates, making their total eligible population a more accurate estimate of the demand for vaccines [8].

Lastly, healthcare and vaccine administration infrastructure also varied among states. Some examples of infrastructure that varied geographically include: the number and size of vaccine administration sites [9], the availability of healthcare personnel [10], and the range of communication platforms used to advertise vaccine eligibility [11].

These are all important factors to account for when orchestrating a mass vaccination campaign. While the effects of hesitancy, infrastructure, and states' VEAPs were difficult to capture in real time, we can examine them retrospectively in order to provide insights for improving the equity and efficiency of future mass-vaccination campaigns. In this paper, we assess the equity and efficiency of the 2021 US COVID-19 vaccine administration campaign by estimating the true supply and demand for vaccines that materialized as a result of states' VEAPs, infrastructure, and population hesitancy. We then estimate the impacts of infrastructure deficits on preventing efficiency and equity of vaccine access across five U.S. states. In doing so, we address the following research objectives, discussed in the Results section:

(O1) Describe how states' unique VEAPs created distinct vaccine demand profiles that impacted equity of vaccine access between states.

(O2) Estimate the individual impacts of hesitancy and infrastructure deficits on the efficiency of the vaccination campaign.

(O3) Identify different sources of infrastructure deficits and examine their relative impact on vaccine administration efficiency.

In Section 5, we discuss the implications of our findings and how these results can inform the design of future mass-vaccination campaigns.

## Literature review

### Equity of vaccine access

There are several different ways to define inequity in the context of health care. Culyer and Wagstaff (1993) claim that these definitions mostly fall on a two-dimensional spectrum of

*horizontal equity*, with people of equal need being treated the same, and *vertical equity*, in which those in greater need should be treated more favorably than those with lesser need [12]. However, others, like Heier Stamm et al. (2017) define and assess equity through a different lens: evaluating systematic disparities in vaccine access between people of different socioeconomic classes and geographic locations [4]. In this paper, we examine the United States' ability to achieve equity of vaccine access between residents in different states, or horizontal equity between states, during the COVID-19 vaccine distribution campaign. We also infer implications in vertical equity by comparing vaccine access within states at different times during the campaign. Since each state's VEAP prioritized vaccine eligibility from high-risk to low-risk individuals as time went on, we capture differences in vertical inequity within states that expanded eligibility at different rates.

By the time COVID-19 vaccines began to be manufactured and distributed, many researchers and government organizations expressed concern regarding the equitable distribution of the vaccines, not only within the United States, but around the world [13–17]. One study revealed massive discrepancies in vaccinated populations between high-income and low-income countries, suggesting that the goal of achieving equity of access was not achieved worldwide [18]. Other studies honed in on equity of access within the United States. Hernandez et. al. (2022) performed a county-level statistical analysis to show that vaccination sites in rural areas were more likely to *not* administer vaccines than sites in metropolitan areas. This trend was also captured in areas with larger versus smaller non-Hispanic black populations, respectively [19].

Other researchers focused on investigating the contributing factors to inequities mentioned in the studies mentioned above. Cardona et al. (2021) defined a vaccination disparity index to highlight how counties with large black and Latinx populations were both disproportionately affected by COVID-19 and had lower vaccination rates due to structural racism and vaccine hesitancy [20]. Howe et al. (2021) conducted a qualitative study of state health department websites to evaluate how the accessibility of eligibility criteria and vaccine appointment scheduling differed between states during the COVID-19 vaccine administration campaign [11]. Finally, Hardeman et al. (2021) examined variations in inclusion criteria of vaccine administration policies and the characteristics of the committees defining the policies of all 50 states and Washington D.C.. The authors suggest that the lack of racial diversity in vaccine equity leadership likely added to the racial inequities observed in vaccine access during the campaign [21]. Similar to Hardeman et al. (2021), we examine details of the vaccine allocation policies enacted by different states. We add onto this work by focusing on the temporal nature of the vaccine policies, highlighting how access and efficiency changed over time. Thus, we measure inequity between states that was caused specifically by states' vaccine allocation plans.

## Vaccine administration efficiency

Much like with equity, there are different lenses through which scientists have studied efficiency of vaccine administration. Goel and Nelson (2021) defined efficiency as the proportion of vaccines administered of those allocated to each state. Using linear regression, the authors showed that states' economic prosperity and rural populations aided COVID-19 vaccine delivery efficiency. States with more nursing homes per capita, health care workers, and COVID-19 deaths were also more likely to have more efficient vaccine delivery [10].

Other researchers viewed vaccine administration efficiency through the lens of immunizing as much of the population as quickly as possible to curb disease spread. A study by Tuite et al. (2021) suggests that given steady vaccine supply, not saving second doses of a 2-dose vaccine regiment could result in 29% fewer COVID-19 cases during a 2-month period. These results

held even when simulating a supply disruption [22]. Along a similar vein, AboulFotouh et al. (2021) addressed the trade-off of using 2-dose vaccines that require cold-storage transportation versus using less effective vaccines that are easier to transport long distances. The authors argue that less effective, un-refrigerated vaccines could be more beneficial than their refrigerated counterparts because of their capability of being transported efficiently, despite having lower efficacy [23].

In this study, we examine inefficiency using Goel and Nelson's (2021) definition, the proportion of vaccines administered of those allocated. However, we also examine efficiency through the lens of eligibility. We define a second measure of efficiency as the proportion of residents vaccinated of those eligible. We show how states' infrastructures, VEAPs, and vaccine allocations led to differences in these two measures of efficiency.

## Vaccine hesitancy

Lastly, we utilize the hesitancy rates that were observed during the COVID-19 pandemic to discern whether states' VEAPs aligned with their hesitancy rates. There are many predictors of vaccines hesitancy that researchers have investigated, including race or ethnicity [8], age [8, 24], social media use [24], government trust [25], and more. We don't account for variation in hesitancy between these groups. Instead, we estimate a constant hesitancy rate, unique to each state. Though certain demographic groups within each state may be more hesitant than others, we conducted our analysis assuming that hesitancy is distributed evenly across all races, ethnicities, ages, occupations, political affiliation, etc.

## Methods

To address the research objectives stated above, we examined vaccine allocation and administration in five US states: California, Massachusetts, Mississippi, New Mexico, and Ohio. These states were selected randomly, and were used in the final analysis because they met two criteria. First, this sample of states has different population densities [26], locations, times of first COVID-19 peak [27], and political leadership in the state legislature [28] in early 2020, shown in Table 1. These attributes make the sample robust to biases associated with geography, political leadership, or size. Second, the selected states all had the entire vaccine-eligibility timeline published on public websites, which was not the case for all US states. Each state had a unique platform for communicating eligibility information and timing. Many of these platforms only showed the most up-to-date information, not the entire eligibility timeline. Thus, for many states, the dates of when certain populations became eligible was no longer available at the time of the analysis.

Each state included in this study had a unique VEAP during the nationwide vaccine administration campaign. All five states began their vaccination campaigns in December 2020 by making vaccines available to healthcare workers and residents of long-term care facilities. In

**Table 1. Characteristics of U.S. states included in this study, collected from [26–28].**

| U.S. State | Population [39] (July 2021) | Pop. Density [39] (2020, Persons/mi$^2$) | State Legislature [28] (2020, Senate + House) | First COVID-19 Peak [9] |
|---|---|---|---|---|
| *California* | 39,142,991 | 253.7 | 90 Dem., 28 Rep. | July 22, 2020 |
| *Massachusetts* | 6,989,690 | 901.2 | 160 Dem., 35 Rep. | April 22, 2020 |
| *Mississippi* | 2,949,586 | 63.1 | 62 Dem., 110 Rep. | July 29, 2020 |
| *New Mexico* | 2,116,677 | 17.5 | 72 Dem., 40 Rep. | May 6, 2020 |
| *Ohio* | 11,764,342 | 288.8 | 47 Dem., 85 Rep. | April 22, 2020 |

the following months, other essential workers, the elderly, and residents with medical conditions became eligible, but the timing and specific subsets of people in these sub-populations differed from state to state. By May 2021, all residents aged 12+ were eligible in all five states, in accordance with a change in federal policy [29].

## Analysis

To assess the equity of vaccine access between the five states **(O1)**, we first examine *competition* for the vaccine within the states. We define "competition" as the discrepancy between eligible residents and vaccines available. When the number of vaccines is lower than the number of eligible residents, we define that there is *competition* for the vaccines. We use competition as a measure of equity both within, and between states because it approximates how difficult it was for any one eligible person to access vaccines in their state. In states with high internal competition, there were many more eligible residents than vaccines available, so it would have been more difficult to find a vaccination site with available vaccines or appointments. On the other hand, eligible residents in states with low internal competition would likely have an easier time finding and scheduling a vaccination appointment. In our analysis, we do not account for vaccine access disparities within states (e.g., geographic disparities) and assume that all residents in a state have equal ability to obtain a vaccine if they want to.

To compare the competition that occurred in different states, we measure competition relative to the total population aged 12+ in each state. In other words, we assess equity of vaccine access by estimating whether some states had more internal competition than others at different times during the campaign.

Additionally, we examine how the internal competition within states changed over time. As previously mentioned, the purpose of the VEAPs was to ensure that the most vulnerable sub-populations would receive vaccinations first. By examining competition over time, we discern if the most vulnerable sub-populations had to compete for vaccines, and how this changed during the campaign.

We calculate competition as a ratio of a state's federal vaccine allocation to their total eligible population each month. However, since some residents were hesitant to get the vaccine in each state, we also compare the vaccines allocated to the number of eligible, non-hesitant residents. This provides a more accurate lens of the competition residents within the same state faced to get the vaccine.

To calculate the total eligible population in each state during every month of the campaign, we use states' VEAPs and state demographic data (discussed below). We define the total eligible population in a state over time as that state's *eligibility profile*. Since this analysis only considers vaccine eligibility up until May 15, 2021, we only include the population of residents aged 12+ in each state, since the vaccine was not yet approved for children aged 0–11 [29]. We define this population as *total residents aged 12+*, and use it to standardize the populations across the different states.

Since new sub-populations became eligible on a frequent basis according to states' VEAPs, we calculated the total eligible population for each state on the 15th day of each month. We calculated the contribution of each sub-population to the total eligible population by assuming that, as sub-populations in different categories became eligible, people in the job-based and health-based categories were distributed proportionally among the age-based categories for people aged 18+. For example, if in February 2020, individuals aged 55–64 became eligible in a state, we assume a proportion of those in that age group were already represented in the health care workers group who became eligible in January 2020.

We then analyzed the efficiency of the vaccine administration campaign **(O2)** by examining the root causes of individuals not getting vaccinated. Each month, each state had a total eligible population. Some of the eligible residents received at least one vaccine dose soon after becoming eligible, while others did not. In this study, we used state hesitancy rates to distinguish which eligible but unvaccinated individuals were (1) unvaccinated due to hesitancy or (2) intended to get vaccinated but did not due to a lack of access or other infrastructure deficits. To calculate the impacts of hesitancy and infrastructure deficits, we first calculated the number of eligible but unvaccinated individuals using states' eligibility profiles and vaccine administration data. Then, we removed the proportion of the total eligible population who we assumed to be hesitant. The remaining, unvaccinated population, we assume, was impacted by infrastructure deficits.

Finally, we examined two drivers of vaccine administration to estimate the relative impacts of infrastructure deficits on vaccine administration efficiency **(O3)**. Particularly, we explored two possible explanations for eligible residents being left unvaccinated each month: (1) state administration efficiency, and (2) vaccine supply versus total eligible demand in each state. Low state administration efficiency would indicate that infrastructure deficits lie in the state administration infrastructure. Phenomena that could prevent peak administration efficiency in states include a lack of available vaccination clinics, poor communication about eligibility or too few health care staff. Otherwise, there is a discrepancy in the supply and demand for vaccines.

We measured vaccine administration efficiency as the proportion of vaccines administered of those allocated to each state [10]. We measured inefficiencies caused by vaccine supply and demand by calculating the ratio of the number of vaccines allocated to the number of vaccines needed to vaccinate the current eligible, non-hesitant population in each state.

## Data collection

In order to calculate the results discussed above, we collected five specific fields of information from each state: (1) the timing and demographic criteria of each state's VEAP, (2) state demographic data, (3) federal vaccine allocation amounts, (4) state vaccine administration counts, and (5) state hesitancy rates.

We gathered information about each state's VEAP via publicly available sources. The eligibility and administration plans for Massachusetts and Ohio were published on their state governments' website [30, 31]. California's was gathered from state and county websites [7, 32]. New Mexico's was published on their Department of Public Health website [33]. Mississippi's was gathered from news sources and Husch Blackwell [6, 34]. Outlines of each state's VEAP can be seen in Table 2.

To estimate the number of eligible residents each month, we utilized demographic data from public sources. States' VEAPs defined eligibility criteria using occupations, health conditions, and age. We used estimates of job- and health-based populations in each state from Ariadne Labs and InfoPlease [35, 36] and used U.S. Census data to estimate age-based populations [26].

Vaccine allocation data was gathered from the CDC, as well as how many of the doses were administered in each state [37]. However, this was not a direct reflection of how many individuals received the vaccine each month since the Pfizer and Moderna vaccines required a second booster dose [38]. Data on the total number of people who received at least one dose of the vaccine was obtained from USA Facts [39].

Finally, we collected data on vaccine hesitancy rates in each state using data from the CDC website [37]. We defined a state's *hesitancy rate* as the proportion of residents aged 12+ who

**Table 2. State's vaccine eligibility and administration plans, gathered from [6, 7, 30–34].**

| State/Date | Populations that became newly eligible in the previous month |
|---|---|
| *California* [5, 31] | |
| Dec. 15, 2020 | Healthcare workers; Residents of long term care facilities. |
| Jan. 15, 2021 | *None.* |
| Feb. 15, 2021 | Residents aged 65+. |
| Mar. 15, 2021 | Workers in emergency services, child care, education, food, and agriculture. |
| Apr. 15, 2021 | Residents aged 50–64. |
| May 15, 2021 | Residents aged 12+. |
| *Massachusetts* [11] | |
| Dec. 15, 2020 | Healthcare workers facing COVID-19. |
| Jan. 15, 2021 | Residents of long-term care facilities; First responders. |
| Feb. 15, 2021 | Remaining healthcare workers; Residents aged 75+. |
| Mar. 15, 2021 | Workers in education and congregate living settings; Residents with 2+ medical conditions; Residents aged 65+. |
| Apr. 15, 2021 | Residents aged 55+. |
| May 15, 2021 | Residents aged 12+. |
| *Mississippi* [23, 41] | |
| Dec. 15, 2020 | Healthcare workers; Residents in long-term care facilities. |
| Jan. 15, 2021 | First responders; Teachers; Other congregate care workers; Residents with pre-existing medical conditions; Residents aged 65+. |
| Feb. 15, 2021 | Residents aged 16–64 with chronic medical conditions |
| Mar. 15, 2021 | Residents aged 12+. |
| Apr. 15, 2021 | *None.* |
| May 15, 2021 | *None.* |
| *New Mexico* [29] | |
| Dec. 15, 2020 | Healthcare workers; Residents of long-term care facilities. |
| Jan. 15, 2021 | aged 75+. |
| Feb. 15, 2021 | Residents aged 16+ with chronic medical conditions. |
| Mar. 15, 2021 | Workers in education. |
| Apr. 15, 2021 | Residents aged 16+. |
| May 15, 2021 | Residents aged 12+. |
| *Ohio* [30] | |
| Dec. 15, 2020 | Healthcare workers; Residents of long-term care facilities. |
| Jan. 15, 2021 | *None.* |
| Feb. 15, 2021 | Workers in education; Residents aged 16+ with pre-existing medical conditions; Residents aged 65+. |
| Mar. 15, 2021 | Workers in law enforcement, child care, or funeral services; Residents under 50 with Type 2 diabetes; Residents aged 16+ with additional medical conditions; Residents aged 50+. |
| Apr. 15, 2021 | Residents aged 16+. |
| May 15, 2021 | Residents aged 12+. |

**Table 3. The hesitancy rate of each state, defined as the percentage of residents aged 12+ still unvaccinated by July 2021, as reported by the CDC [37].**

| U.S. State | Hesitancy Rate |
|---|---|
| *California* | 23.6% |
| *Massachusetts* | 16.1% |
| *Mississippi* | 52.6% |
| *New Mexico* | 22.4% |
| *Ohio* | 41.4% |

were still unvaccinated after July 2021. After July 2021, vaccination rates had plateaued in all five states and all residents who intended to get vaccinated should have been able to in the three months prior. Additionally, residents aged 0–11 were still not eligible to receive the vaccine at this time. We assumed that any residents still unvaccinated by July 2021 never intended to get vaccinated as they became eligible during the campaign. Due to a lack of more granular data, we assumed that hesitancy was evenly distributed among all of the sub-populations in each state and that hesitancy remained unchanged over time. The resulting hesitancy rates are shown in Table 3.

## Results

### Equity of vaccine access

Differences in states' VEAPs resulted in the states having different eligibility profiles. Fig 1a displays these eligibility profiles for the five states examined in this study. Certain states, like Mississippi, had a larger proportion of their population eligible for the vaccine sooner than others. Mississippi structured their VEAP such that the majority of the state's population (≈60%) was eligible by mid-January 2021. Mississippi was also the first of the five states to open vaccine eligibility to the entire population aged 12+ in March 2021. Mississippi exhibited a *front-heavy* approach to vaccine eligibility, opening eligibility to more residents quickly. On the other hand, California, Massachusetts, New Mexico, and Ohio all exhibited *back-heavy* approaches, where the majority of the states' populations were not eligible until at least April 2021.

Fig 1b shows the cumulative number of vaccines allocated to each state relative to the total number needed to fully vaccinate the entire population aged 12+. The similar trajectory of the

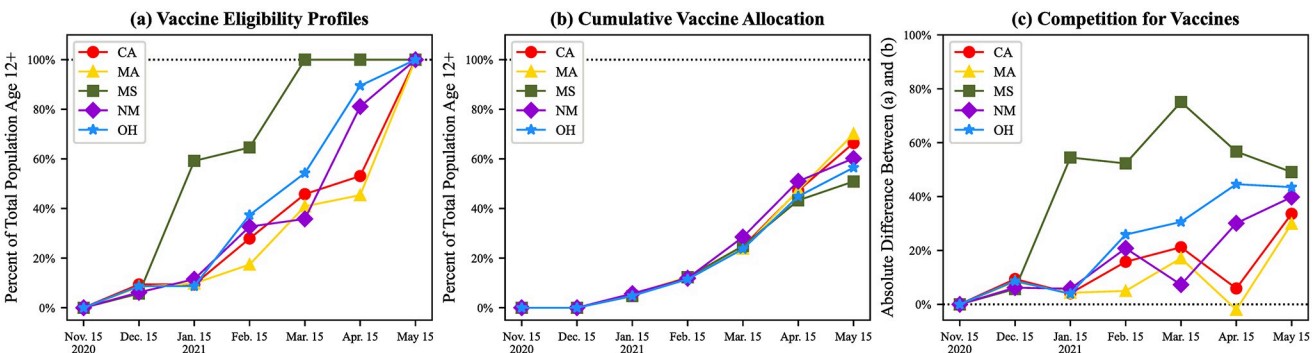

**Fig 1. (a) Vaccine eligibility profiles of the 5 states, (b) cumulative number of vaccines allocated each month, and (c) discrepancies between total eligible populations and cumulative number of vaccines allocated to each state.**

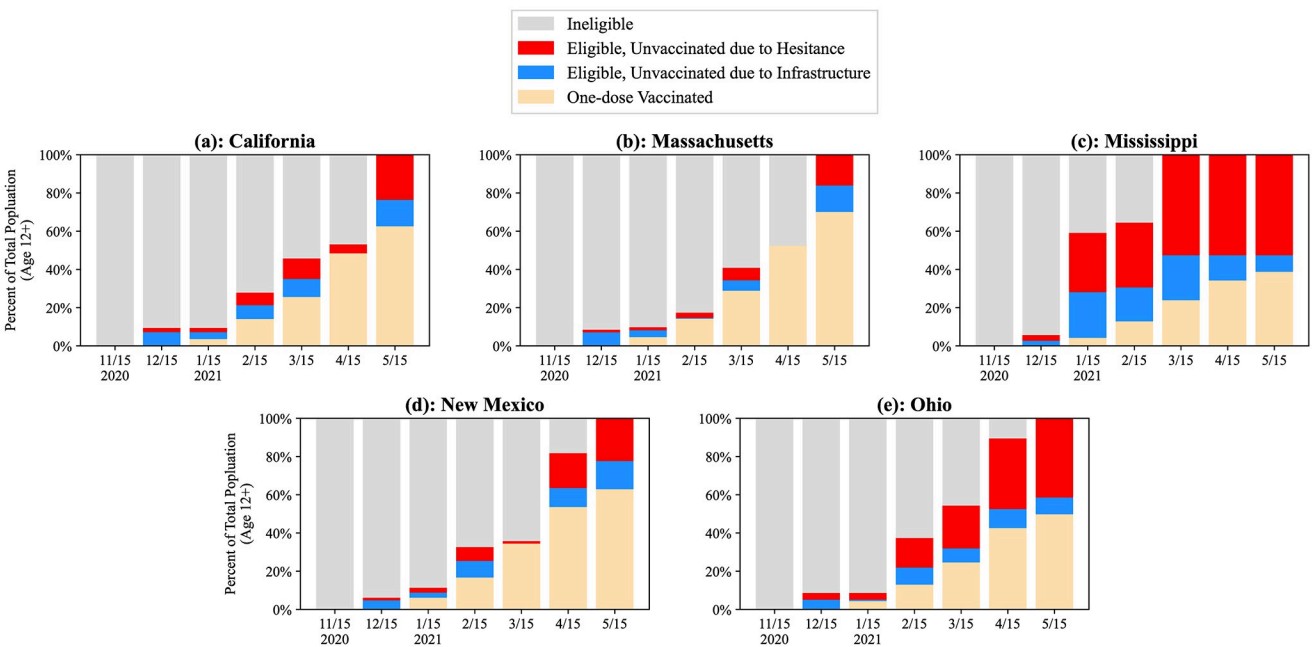

**Fig 2. Estimates of the percentage of each state's population aged 12+ who were either ineligible (grey), eligible but unvaccinated due to hesitancy (red), eligible but unvaccinated due to infrastructure deficits (blue), or vaccinated with at least one-dose vaccinated (tan), during each month of the vaccine administration campaign.**

traces in Fig 1b suggests that the federal government allocated vaccines to states proportionally to their population throughout the campaign [40]. Since vaccines were allocated this way, the differences in eligibility profiles resulted in different competition within each of the states. Fig 1c shows a measurement of competition within states for vaccines, that is, the relative difference between the eligible population and the vaccine allocation. In states with more *front-heavy* VEAPs, like Mississippi, there was more competition in earlier months to receive the limited vaccines available in that state. During some months in Mississippi, there were as many as 50% more citizens eligible than vaccines available to fully inoculate them. While, in other states, there was only 10–30% more eligible individuals. The four states that exhibited *back-heavy* VEAPs all had similar competition for vaccines throughout the campaign. Competition for vaccines grew as time went on and as more residents became eligible in each state, suggesting that there was less competition for more vulnerable populations than for lower-risk populations.

## Vaccine administration inefficiency

Given their vaccine allocations from the federal government, each state attempted to vaccinate their residents as quickly as possible according to their VEAP. We examined the vaccine administration in each state with a focus on quantifying the impact of hesitancy and infrastructure on the supply and demand for vaccines. Fig 2a–2e show the proportion of each state that was vaccinated with at least one dose (tan), eligible but not vaccinated (union of blue and red), and not yet eligible (grey) according to the vaccine administration data gathered from USA Facts [39]. In almost all instances, there were large eligible populations that did not get vaccinated despite being eligible (union of red and blue in Fig 2).

Residents who were unvaccinated each month were assumed to either have chosen not to be vaccinated or were unable to get vaccinated due to an infrastructure deficit (e.g. insufficient vaccine supply, lack of knowledge that they were eligible, lack of access to vaccine clinics, etc.). By also assuming that hesitancy was evenly distributed among all of the sub-populations defined in the VEAPs, we separated eligible, unvaccinated individuals into two categories: (1) people who were hesitant and likely would never get vaccinated (red), and (2) people who were unvaccinated due to infrastructure deficits (blue). In each of the five states studied, there were significant proportions of the populations each month that were left unvaccinated despite being eligible and not hesitant. These infrastructure deficits were most pronounced in Mississippi, the state which had the most *front-heavy* VEAP, despite also having the highest hesitancy rate. In January 2021, Mississippi had, 23.9% of its residents eligible but unable to get vaccinated within the month due to infrastructure deficits. These results demonstrate that the major inefficiencies in vaccinating eligible individuals cannot be fully accounted for by hesitant populations, and infrastructure deficits should be considered.

## Sources of infrastructure deficits

Beyond demonstrating the toll of infrastructure deficits, we further investigated the source of these deficits. To do this, we isolated the roles played by the vaccine supply versus eligible demand and state vaccination efficiency during the campaign.

We first examined the vaccine administration efficiency in each state. Fig 3 shows the percentage of the total allocated vaccines that were administered each month. We observe that by February 2021, all five states established infrastructure to consistently administer >65% of vaccines allocated to them. Additionally, vaccine administration efficiency remained above 75% for all states except Mississippi in later months, despite the increased allocations from the federal government. Based on these findings, from February 2021 on, the states could have

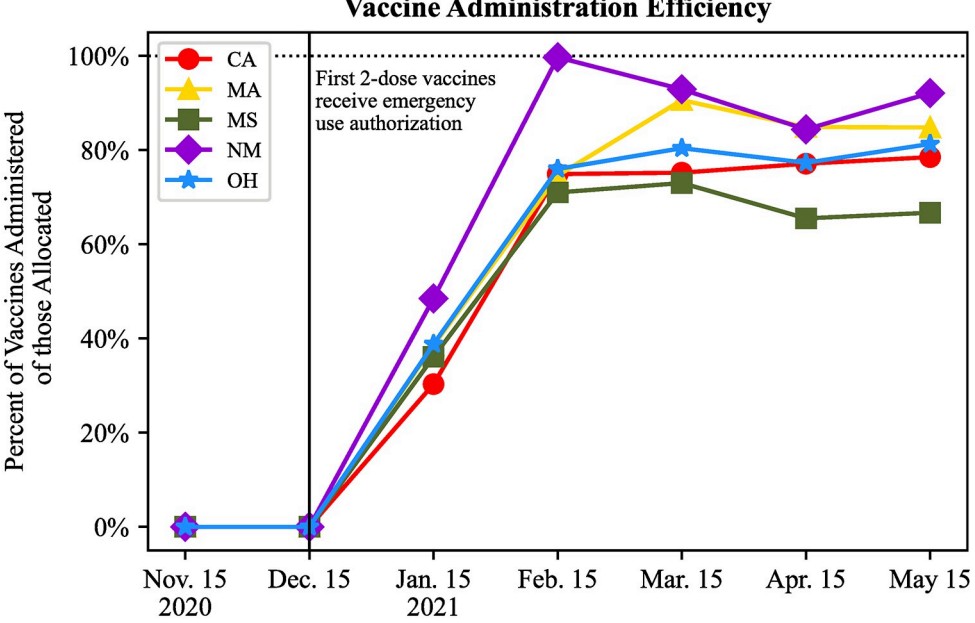

**Fig 3. Vaccine administration efficiency of each state relative to their allocation.**

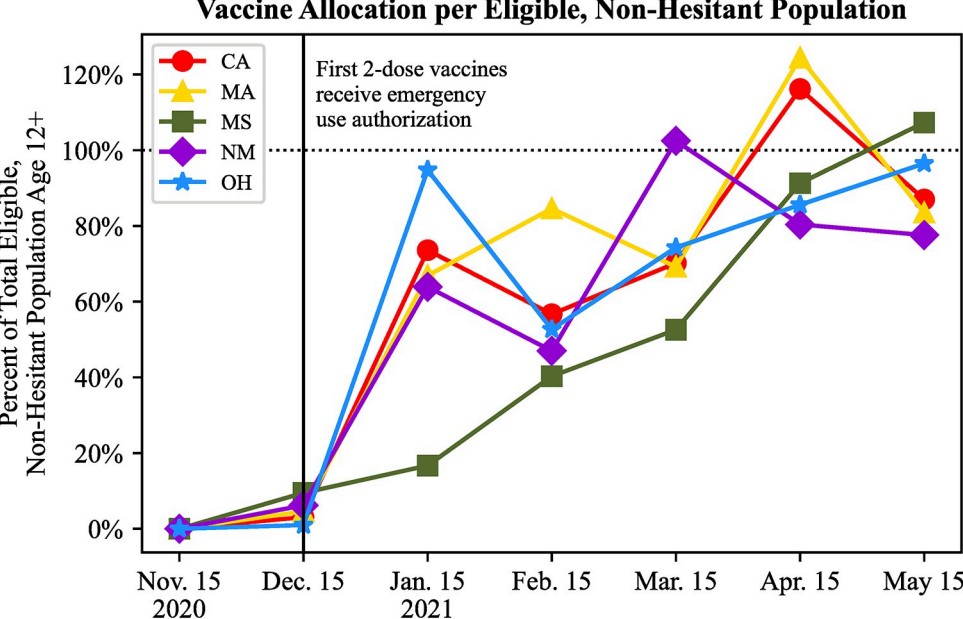

**Fig 4. Number of vaccines allocated to each state relative to the total vaccines needed to fully vaccinate the eligible, non-hesitant population each month.**

increased vaccination administration efficiency by at most 35% through improving their administration infrastructure.

Though state infrastructure is a contributor to the overall inefficiency of the vaccination campaign, it does not account for the all of the unvaccinated populations seen each month in Fig 2a–2e. Instead, the disparity between supply and demand of vaccines was largely due to a mismatch between federal vaccine supply and total eligible demand within states, especially in the early months of the vaccine administration campaign. Fig 4 shows the ratio of vaccines allocated from the federal government to each state as compared to the non-hesitant, eligible population in each month. Note that for most states in most months, there were not as many vaccines allocated to the state as were needed to fully vaccinate the eligible, non-hesitant residents. In other instances, more vaccines were allocated to the state than was needed to vaccinate the eligible, non-hesitant population (e.g. Massachusetts and California in April 2021). Not only was there insufficient supply to match the demand for vaccines in the early months of the campaign, but there was also excess vaccines allocated to some states in the later months. These results demonstrate a lack of coordination between vaccine allocations and states' VEAPs driving the inefficiency of vaccinating eligible residents.

## Discussion

This study demonstrates how the variations in states' VEAPs created inequity in vaccine access between the states. States with *front-heavy* VEAPs, such as Mississippi, had more internal competition for vaccines than other states. Even when accounting for hesitancy in the population, Mississippi still had more unvaccinated residents due to infrastructure than the other four states throughout the entire campaign. Despite the federal government's attempts to achieve equity in vaccine distribution by allocating vaccines proportionally to state populations, the

differences in eligibility timing between the states created differences in accessibility for residents.

Additionally, we show that there were discrepancies between the supply of vaccines and the total eligible demand for vaccines in all five states during the entire course of the campaign. In most instances, there were more eligible, non-hesitant residents each month than there were vaccines to fully inoculate them. Thus, there were some systematic factors that were holding back the campaign from being maximally efficient. By examining two sources of infrastructure deficits, we show that both state infrastructure and federal vaccine supply contributed to the inefficiency of the vaccine administration campaign; neither were fully to blame.

Though all five states in this study did not achieve 100% efficiency when administering their allocated vaccines, 4 of 5 achieved >75% efficiency from February 2021 onward, keeping up with the increased allocations every month. Even if the states had achieved maximum efficiency, the federal allocations were still not large enough to vaccinate all eligible residents at that time. So, the discrepancy between federal vaccine allocations and eligible demand between the states was the primary driver in vaccine delivery inefficiency at each of the five states.

Discrepancies between the supply and demand for vaccines has the potential to diminish the safety of the vaccine administration campaign. With more people eligible than vaccines available, some residents may obtain the vaccine before other, more at-risk residents. In an ideal scenario, vaccines should be administered to the most vulnerable individuals first (i.e. establishing vertical equity), but this may not occur if significant proportions of a state's population are competing for few vaccines.

To fix this discrepancy and increase the safety and efficiency of the campaign, either the production of vaccines would need to increase to meet demand deemed eligible by the VEAPs, or the VEAPs would need to change to accommodate the limited supply of vaccines. Increasing vaccine production may seem ideal, but it would be extremely costly and constrained by production capacities of the companies producing the vaccine. Major changes in production infrastructure would also not be practical in the time span of the vaccine roll out. And ultimately, limited manufacturing capacity is largely why VEAPs prioritizing high-risk populations need to be created in the first place.

Instead of increasing vaccine production, VEAPs could be changed to more closely match the allocation of vaccines each month. By decreasing the number of eligible residents each month, there would be less competition for the vaccines, increasing the overall safety of the distribution process by ensuring high-risk residents receive vaccines early. However, having excess demand, as seen in this case, minimizes the likelihood of any left-over, unused vaccines that could be used elsewhere. So, there is a balance to be obtained between opening slower to decrease competition and opening faster to ensure vaccines are being administered as quickly as possible.

Additionally, it is difficult to design VEAPs when the true demand for vaccines and size of future vaccine allocations are uncertain. So, new VEAPs should be designed to take into consideration up-to-date information on vaccine hesitancy, vaccination rates, and future allocations to decide how best to proceed with opening eligibility to additional residents. An adaptive approach to VEAPs, instead of one that is set in stone well in advance, is a method that should be considered for future mass vaccination campaigns. Such a design could incentivize vaccine administration efficiency in states while still prioritizing the most high-risk individuals. Alternatively, a centralized inventory management system could be used to address inequity in real time by transferring vaccines from states with excess supply to states with limited supply.

## Limitations

Two primary limitations of this work include (1) the use of a small sample of states and (2) the use of decentralized data sources to construct VEAPs and eligible population profiles for each state. This study only examines the efficiency and equity of vaccine distribution in 5 of the 50 US states. This small sample size was chosen partially due to the lack of available information on states' eligibility criteria and also the decentralized nature of this data. Thus, we are careful about drawing broad conclusions about the overall success of states' vaccine administration campaigns. However, considering how consistent the discrepancy between vaccine supply and true demand were in each state, and the variety of states included in this study, it is unlikely that the small sample set of states are hindering the generalizability of these results. Since, during our initial selection of states, none of the 50 states stood out as having slow eligibility plans, that would more closely match federal vaccine supply, our results would be unlikely to change by examining additional states.

As previously mentioned, this study collected data from several public, non-government sources to calculate the results. Thus, the accuracy of our results is dependent on the accuracy of the data sources used and our assumptions about constant population size and uniform dispersion of hesitancy. Additionally, the use of decentralized sources limits the comparability of results between states. However, the sources used to find the total eligible populations, sizes of sub-populations, vaccines allocated, and vaccines administered were all collected from the same sources for each of the five states, which are the primary data that the results rely on. Thus, the accuracy and generalizability of results shown in this study is strictly dependent on the accuracy of the individual sources. This study's reliance on decentralized, non-government sources points to the need for improved data collection methods during future vaccine administration campaigns so that research reflecting on the success of the campaigns can be as accurate as possible.

## Conclusion

In this study, we utilized vaccine allocation and administration data to explore how states' vaccine eligibility and administration plans and vaccine hesitancy impacted the equity and efficiency of vaccine administration during COVID-19. Expanding on previous work, we examine states' VEAPs temporally, allowing us to measure equity and efficiency at different time points in the campaign. We show that because of limited vaccine availability in early 2021, states that granted eligibility to more residents sooner created more competition among residents to receive vaccines than other states. Not only did differences in states' VEAPs lead to inequity of access throughout the campaign, but those differences also explain why, in many instances, significantly more residents were eligible than there were vaccines to inoculate them. The lack of coordination between federal vaccine allocations and state's VEAPs led to inefficiencies in vaccinating residents throughout the campaign. To achieve a more efficient vaccination campaign that ensures vulnerable residents are not competing for vaccines, an adaptive approach to expanding eligibility should be considered. Such an approach could account for vaccine allocations, hesitancy, and vaccination efficiency in real time to orchestrate a more efficient and equitable campaign.

### Future work

There are a variety of mathematical models and simulation tools that could be developed to inform the design of such an adaptive vaccine allocation system. In this study, we have highlighted key parameters that should be incorporated into this system including vaccine hesitancy, sub-population sizes, total eligible demand, and vulnerability. There is a need to utilize

these concepts to develop models of a nation-wide vaccine administration campaign with the goal of maximizing efficiency, equity, and safety in real time. Such models can build on vaccine allocation models such as those developed by Balcik et al. [41] and Dastgonshade et al. [42]. These models could then be adapted to incorporate real data from multiple states to help orchestrate future mass-vaccination campaigns.

## Supporting information

**S1 File. Excel file containing the collected data and calculations done for the results shown in this study.** All references, data, formulas, etc. are shown for each individual graph in a separate sheet.
(XLSX)

## Author Contributions

**Conceptualization:** Noah Chicoine, Noah Schnipper, Jacqueline Griffin.

**Data curation:** Noah Chicoine, Noah Schnipper.

**Formal analysis:** Noah Chicoine, Noah Schnipper.

**Methodology:** Noah Chicoine, Noah Schnipper, Jacqueline Griffin.

**Project administration:** Noah Chicoine, Jacqueline Griffin.

**Supervision:** Jacqueline Griffin.

**Visualization:** Noah Chicoine.

**Writing – original draft:** Noah Chicoine, Noah Schnipper.

**Writing – review & editing:** Noah Chicoine, Jacqueline Griffin.

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
