## [Decision Letter · Decision Letter 0]

8 Mar 2024

PONE-D-23-33166The Role of Hesitancy and Infrastructure in the Equity and Efficiency of COVID-19 Vaccine AdministrationPLOS ONE

Dear Dr. Chicoine,

Thank you for submitting your manuscript to PLOS ONE. After careful consideration, we feel that it has merit but does not fully meet PLOS ONE’s publication criteria as it currently stands. Therefore, we invite you to submit a revised version of the manuscript that addresses the points raised during the review process.

The manuscript is well structured and scientifically valid, however, some changes are needed as indicated by reviewers.

We look forward to receiving your revised manuscript.

Kind regards,

Andrea Cioffi

Academic Editor

PLOS ONE

Journal Requirements:

Reviewers' comments:

Reviewer's Responses to Questions

**Comments to the Author**

1. Is the manuscript technically sound, and do the data support the conclusions?

Reviewer #1: Partly

Reviewer #2: Yes

2. Has the statistical analysis been performed appropriately and rigorously? 

Reviewer #1: Yes

Reviewer #2: Yes

3. Have the authors made all data underlying the findings in their manuscript fully available?

Reviewer #1: Yes

Reviewer #2: Yes

4. Is the manuscript presented in an intelligible fashion and written in standard English?

Reviewer #1: Yes

Reviewer #2: Yes

5. Review Comments to the Author

Reviewer #1: This is an interesting article that I enjoyed reading. It is well written and deals with an important and much debated topic. I have a few comments/suggestions

- there is some repetitiveness in the introduction and literature review. The research questions are stated at the end of the introduction so the aims don’t need to be mentioned again in the first paragraph of the literature review, and if they do, then they have not been explained well enough.

- lines 196-250 has question marks for the figure numbers which need removing.

- can you explain more fully why the 5 states included were chosen? Was this random or due to data availability or other criteria?

- please explain in the limitations section that the sources of data used were very different and therefore not generalisable e.g. media was used for one

- in the future work section you mention the use of models - do any of these already exist?

Reviewer #2: This study reviewed five states’ VEAPs, federal vaccine allocations, vaccine administrations, and vaccine hesitancy to assess the equity of vaccine access and vaccine administration efficiency that manifested

during the COVID-19 vaccine campaign. The manuscript has many strengths, including the introduction and literature review. The information presented was clear and concise. The analysis was also very strong, with a thorough description of how equity, efficiency, and hesitance was assessed. There were two grammatical errors/typos (line 184, line 207). The manuscript implications section was especially interesting, as it provided suggestions for ensuring equitable and efficient distribution of vaccines or medical resources during health crises.

6. PLOS authors have the option to publish the peer review history of their article (what does this mean?). If published, this will include your full peer review and any attached files.

Reviewer #1: **Yes: **Dr Bethan Swift

Reviewer #2: No

---

## [Author Response · Author response to Decision Letter 0]

12 Apr 2024

Please see the Response to Reviewers document attached in this submission.

---

## [Decision Letter · Decision Letter 1]

13 May 2024

The Role of Hesitancy and Infrastructure in the Equity and Efficiency of COVID-19 Vaccine Administration

PONE-D-23-33166R1

Dear Dr. Chicoine,

We’re pleased to inform you that your manuscript has been judged scientifically suitable for publication and will be formally accepted for publication once it meets all outstanding technical requirements.

Kind regards,

Andrea Cioffi

Academic Editor

PLOS ONE

Additional Editor Comments (optional):

Reviewers' comments:

Reviewer's Responses to Questions

**Comments to the Author**

1. If the authors have adequately addressed your comments raised in a previous round of review and you feel that this manuscript is now acceptable for publication, you may indicate that here to bypass the “Comments to the Author” section, enter your conflict of interest statement in the “Confidential to Editor” section, and submit your "Accept" recommendation.

Reviewer #2: All comments have been addressed

2. Is the manuscript technically sound, and do the data support the conclusions?

Reviewer #2: Yes

3. Has the statistical analysis been performed appropriately and rigorously? 

Reviewer #2: Yes

4. Have the authors made all data underlying the findings in their manuscript fully available?

Reviewer #2: Yes

5. Is the manuscript presented in an intelligible fashion and written in standard English?

Reviewer #2: Yes

6. Review Comments to the Author

Reviewer #2: (No Response)

7. PLOS authors have the option to publish the peer review history of their article (what does this mean?). If published, this will include your full peer review and any attached files.

Reviewer #2: **Yes: **Dr. Nenette A. Cáceres

---

## [Editor Report · Acceptance letter]

4 Jun 2024

PONE-D-23-33166R1 

PLOS ONE

Dear Dr. Chicoine, 

I'm pleased to inform you that your manuscript has been deemed suitable for publication in PLOS ONE. Congratulations! Your manuscript is now being handed over to our production team.

Kind regards, 

on behalf of

Dr. Andrea Cioffi 

Academic Editor

PLOS ONE